# Cattle Manure Fertilizer and Biostimulant *Trichoderma* Application to Mitigate Salinity Stress in Green Maize Under an Agroecological System in the Brazilian Semiarid Region

**DOI:** 10.3390/plants14233643

**Published:** 2025-11-29

**Authors:** Maria Vanessa Pires de Souza, Geocleber Gomes de Sousa, Arthur Prudêncio de Araujo Pereira, Henderson Castelo Sousa, Kaio Gráculo Vieira Garcia, Leonardo Vieira de Sousa, Gerônimo Ferreira da Silva, Ênio Farias de França e Silva, Thieres George Freire da Silva, Alexsandro Oliveira da Silva

**Affiliations:** 1Center of Agrarian Sciences, Federal University of Ceará (UFC), Av. Mr. Hull 2877, Fortaleza 60356-001, CE, Brazil; castelohenderson@gmail.com (H.C.S.); kaiovieira@ufc.br (K.G.V.G.); 2Institute of Rural Development, University of International Integration of Afro-Brazilian Lusofonia (UNILAB), Redenção 62790-000, CE, Brazil; sousagg@unilab.edu.br (G.G.d.S.); leoigt@hotmail.com (L.V.d.S.); 3Department of Agricultural Engineering, Federal Rural University of Pernambuco (UFRPE), Dois Irmãos, Recife 52171-900, PE, Brazil; geronimo.silva@ufrpe.br (G.F.d.S.); enio.fsilva@ufrpe.br (Ê.F.d.F.e.S.); thieres.silva@ufrpe.br (T.G.F.d.S.); alexsandro.osilva@ufrpe.br (A.O.d.S.)

**Keywords:** *Zea mays* L., *Trichoderma harzianum*, cattle manure

## Abstract

Organic fertilizer stimulates soil microbial activity, contributing to the decomposition of organic residues and nutrient release, thereby helping to reduce the negative effects of soil salinity. The use of *Trichoderma* can enhance nutrient availability to plants and increase their resistance to abiotic stresses, such as salinity stress. Therefore, this study aimed to determine the effects of organic fertilizer on green maize plants under soil inoculation with *Trichoderma harzianum* and salinity stress. The experiment was conducted during the 2022 and 2023 growing seasons in Redenção municipally, Ceará, Brazil. A randomized complete block design was used, arranged in a split–split plot scheme with four replications. The main plots consisted of two electrical conductivities of irrigation water (ECw): A1—0.3 dS m^−1^ and A2—3.0 dS m^−1^. The subplots comprised three doses of well-composted cattle manure (0, 10, and 20 t ha^−1^), and the sub-subplots represented soil inoculation (TI) or non-inoculation (NI) with the *Trichoderma harzianum*-based biostimulant. Irrigation with saline water reduced maize growth and yield; however, these effects were mitigated by *Trichoderma harzianum* and organic fertilizer. Inoculation was particularly effective under lower electrical conductivity conditions. The combination of 10 t ha^−1^ of cattle manure with *Trichoderma* inoculation showed the highest productive efficiency.

## 1. Introduction

Maize (*Zea mays* L.) is an essential global crop, playing a crucial role in both human and animal nutrition [1]. Its successful cultivation depends on factors such as favorable climatic conditions, soil characteristics, and effective management practices [2]. However, challenges such as the efficient management of water resources, the need for more sustainable agricultural practices, and the increasing salt concentration in irrigation water can compromise the agronomic performance of green maize [3,4].

The use of brackish water for irrigation has been adopted in arid and semi-arid regions worldwide as a strategy to expand agricultural production in response to freshwater scarcity [5]. Globally, localized irrigation with moderately saline water (up to 3 dS m^−1^) has enabled the cultivation of various crops, although with reduced yields due to the detrimental effects of salts on plants [6]. The use of salt-tolerant species and adapted management systems, such as drip irrigation and preventive drainage, has mitigated the negative impacts of salinity on plants and prevented soil salinization [7].

High salt concentrations in the soil are common in irrigated areas, where excessive salt accumulation in the root zone hinders water uptake by plant roots, particularly in arid and semiarid regions characterized by high evapotranspiration rates and irregular rainfall [8]. Salinity stress leads to nutrient imbalances, alterations in osmotic potential, ionic toxicity, and reduced crop productivity [9,10].

Maize has been classified by [11] as a moderately salt-sensitive crop, with threshold values of approximately 1.7 dS m^−1^ for soil electrical conductivity and 1.1 dS m^−1^ for irrigation water. When these limits are exceeded, plants often exhibit reduced growth [12], declines in photosynthetic efficiency [13], and decreases in yield [14].

Thus, the use of organic fertilizer stands out as a sustainable strategy to improve soil fertility and enhance the productivity of cultivated plants [15]. In addition to supplying essential nutrients, organic fertilizer stimulates soil microbial activity, contributing to the decomposition of organic residues and the release of nutrients such as nitrogen, phosphorus, potassium, and sulfur to plants [16]. This process not only increases crop productivity but also reduces the concentration of soluble salts, facilitating water absorption by roots and mitigating the negative effects of salinity on photosynthetic processes [14,17].

The growing demand for biological solutions in agriculture reflects the search for efficient microorganisms that promote both economic and environmental sustainability [18]. Among the microorganisms studied, the *Trichoderma* genus stands out for its ability to establish symbiotic associations with plant roots, improving nutrient uptake and enhancing plant resistance to environmental stresses such as salinity and nutrient deficiency [19]. Studies focusing on the use of *Trichoderma* in combination with different fertilizer rates and irrigation with brackish water are still scarce in the literature. Therefore, this research aimed to determine the effects of organic fertilizer and soil inoculation with *Trichoderma harzianum* on the production of green maize subjected to salinity stress.

## 2. Results

As shown in Table 1, during the first cultivation cycle, an interaction was observed between fertilizer rates (FR) and inoculation (I) for plant height (PH), leaf area (LA), stem diameter (SD), and number of leaves (NL), as was an interaction between irrigation water electrical conductivity (ECw) and inoculation (I) for leaf area (LA). In the second cycle, a triple interaction among ECw, fertilizer rates, and inoculation was detected for plant height (PH) and stem diameter (SD). The variable leaf area (LA) showed a double interaction, both between ECw and inoculation (I), and between fertilizer rates (FR) and inoculation (I).

In the first cultivation cycle, a significant interaction was observed between fertilizer rates in treatments with and without *Trichoderma* inoculation (Figure 1A). The dose of 20 t ha^−1^ resulted in greater plant height (107.68 cm), not differing statistically from the 10 t ha^−1^ dose in the non-inoculated (NI) treatments. However, the inoculation with *Trichoderma* (TI) at the 10 t ha^−1^ dose provided the greatest plant height (112.5 cm) (Figure 1A). In the second cultivation cycle (Figure 1B), a triple interaction among treatments was observed. The greatest plant heights (90 and 87.5 cm) were obtained at the lowest salinity level combined with the 10 and 20 t ha^−1^ fertilizer rates in the presence of *Trichoderma*.

Figure 2A,B show that in both the first and second cultivation cycles, the inoculated treatment exhibited a significantly greater leaf area under the lowest salinity level, with mean values of 380.68 cm^2^ and 500.5 cm^2^, respectively. At the highest salinity level, however, the mean values of the non-inoculated and inoculated treatments did not differ significantly. As shown in Figure 2C, the 10 t ha^−1^ dose resulted in the greatest leaf area (403.16 cm^2^) during the first cultivation cycle in the *Trichoderma*-inoculated (TI) treatments. In the second cultivation cycle (Figure 2D), within the TI treatments, the 10 and 20 t ha^−1^ doses contributed to larger leaf areas, with mean values of 484.16 cm^2^ and 486.24 cm^2^, respectively. The 10 t ha^−1^ dose resulted in a greater leaf area in maize plants irrigated with low-salinity water (Figure 2E). In this case, organic fertilizer with 10 t ha^−1^ was the most efficient in promoting leaf area expansion.

As shown in Figure 3A, no significant differences were observed among fertilizer rates (FD) in the non-inoculated (NI) treatments. However, in the *Trichoderma*-inoculated (TI) treatment, the 10 t ha^−1^ dose resulted in the greatest stem diameter, while the 20 t ha^−1^ dose did not provide additional benefits during the first cultivation cycle. These results suggest that the combination of *Trichoderma* with appropriate organic fertilizer may be an effective strategy to maximize stem diameter.

In the second cultivation cycle (Figure 3B), inoculation with *Trichoderma* (TI) resulted in a greater stem diameter compared with non-inoculated (NI) treatments at an ECw of 0.3 dS m^−1^ for the 10 and 20 t ha^−1^ doses. At an ECw of 3.0 dS m^−1^, this positive effect was observed only in the absence of fertilizer.

For the variable number of leaves (Figure 4), no significant differences were observed among fertilizer rates in soil without *Trichoderma* inoculation. However, in soil inoculated with the microorganism (TI), the 10 t ha^−1^ dose resulted in a higher number of leaves, though not significantly different from the 0 t ha^−1^ dose.

As shown in Table 2, in the first cultivation cycle, a triple interaction was observed among irrigation water electrical conductivity (ECw), organic fertilizer rates (FR), and *Trichoderma harzianum* inoculation for the variables stomatal conductance (gs) and chlorophyll index (SPAD). For the transpiration rate, a double interaction between ECw and FR was observed. The CO_2_ assimilation rate (A) was influenced independently by ECw and inoculation. In the second cycle, the triple interaction among ECw, FR, and inoculation was significant only for stomatal conductance (gs), whereas the CO_2_ assimilation rate (A) showed an interaction between ECw and inoculation.

In Figure 5A, it can be observed that irrigation water with a higher salinity level (3 dS m^−1^) resulted in a greater CO_2_ assimilation rate compared with plants irrigated with lower-salinity water (0.3 dS m^−1^) during the first cultivation cycle. In the results from the second cultivation cycle (Figure 5B), no significant differences were observed between the NI and IT treatments at the lower salinity level (0.3 dS m^−1^). However, at the higher salinity level (3 dS m^−1^), the NI treatment showed a greater CO_2_ assimilation rate. In Figure 5C, the presence of *Trichoderma* resulted in a higher CO_2_ assimilation rate compared with plants that were not inoculated with the fungus.

No significant differences were observed among fertilizer rate within each salinity level for the transpiration variable (Figure 6).

For the variable gs (Figure 7A), a significant interaction among the factors was observed. At the lower salinity level, gs was not affected by inoculation (I) or fertilizer rates (0 and 10 t ha^−1^), indicating that the bioinoculant did not influence this variable under these conditions. However, at the 20 t ha^−1^ dose, stomatal conductance increased significantly in the presence of the microorganism. At the higher salinity level, only the 0 t ha^−1^ dose showed a significant difference in stomatal conductance, with an increase observed in the absence of the microorganism (NI).

In the second cultivation cycle (Figure 7B), under the lower salinity level (0.3 dS m^−1^), no significant differences were observed among fertilizer rates for either inoculated (TI) or non-inoculated (NI) treatments, except at the 20 t ha^−1^ dose, where the inoculated treatment (TI) exhibited significantly higher stomatal conductance. This result may be associated with the greater availability of organic matter, which, combined with the action of *Trichoderma*, favored root development and stomatal activity. At the higher salinity level (3.0 dS m^−1^), treatments inoculated with *Trichoderma* (TI) showed higher stomatal conductance values at the 0 and 10 t ha^−1^ doses. In contrast, at the 20 t ha^−1^ dose, the non-inoculated treatment (NI) exhibited higher stomatal conductance.

In Figure 8, it was observed that treatments inoculated with *Trichoderma* (TI) showed a significant increase in the chlorophyll index. Under lower salinity conditions (0.3 dS m^−1^), the 0 and 10 t ha^−1^ rates exhibited higher chlorophyll index values in the TI treatment. Under higher salinity (3.0 dS m^−1^), only the 0 t ha^−1^ dose differed significantly, showing a higher chlorophyll index in the TI treatment. These results suggest that *Trichoderma* may act as a biostimulant, as it demonstrated a positive effect by increasing the chlorophyll index under both irrigation water conductivity levels evaluated.

The results presented in Table 3 indicate that, in the first cultivation cycle, there was an interaction between organic fertilizer rates and *Trichoderma harzianum* inoculation for the variables ear yield without husk (EYWH) and ear yield with husk (EYH), as well as an interaction between irrigation water electrical conductivity (ECw) and inoculation (I) for EYH. An isolated effect of inoculation (I) was observed for ear diameter (ED), and an isolated effect of fertilizer rate (FD) for ear length (EL). In the second cultivation cycle, an isolated effect of inoculation (I) was observed for the variable ear yield with husk (EYH).

The 10 t ha^−1^ dose of cattle manure promoted the greatest ear length (Figure 9A). For ear diameter (Figure 9B), the treatments inoculated with *Trichoderma* (TI) resulted in greater ear diameter values. These results reinforce the effectiveness of the biostimulant in enhancing the plant’s growth capacity.

For the variable number of grains per row (Figure 10A), there was an interaction effect between inoculation and fertiliser doses. The presence of *Trichoderma* positively influenced the number of grains for doses 0 and 10 t ha^−1^, although the latter was not statistically significant. For the variable number of rows per ear (Figure 10B), there was an isolated effect for fertiliser doses, with the 0 t ha^−1^ dose presenting the highest mean (12.90), while the 10 and 20 t ha^−1^ doses did not differ significantly from each other, with means of 11.70 and 11.56, respectively.

Ear yield with husk (EYH) was significantly influenced by *Trichoderma* inoculation when plants were irrigated with low-salinity water, showing higher values compared with the non-inoculated (NI) treatment. However, under irrigation with high-salinity water, no significant differences were observed between NI and TI treatments during the first cultivation cycle (Figure 11A). In the second cultivation cycle (Figure 11B), there was an isolated inoculation effect. The non-inoculated treatment (NI) showed a higher ear yield with husk (EYH) compared with the treatment inoculated with *Trichoderma harzianum* (TI). This result may be related to several factors, one of which is the continuous use of brackish water. Although not statistically significant for this variable, prolonged exposure may have affected the colonization and symbiotic activity of *Trichoderma* in the plant roots.

Fertilizer rates influenced ear yield with husk (EYH) during the first cultivation cycle (Figure 11C). In the presence of *Trichoderma* (TI), the 10 t ha^−1^ dose resulted in higher EYH compared with the 20 t ha^−1^ dose. However, no significant differences were observed between the 0 and 10 t ha^−1^ doses. In Figure 11D, which also refers to the first cycle, it can be observed that treatment with *Trichoderma* (IT) at a dose of 10 t ha^−1^ resulted in higher productivity of ears without straw (PESP) compared to the dose of 20 t ha^−1^. However, no statistically significant differences were found between the doses of 0 and 10 t ha^−1^, indicating that inoculation with T. harzianum was effective in maintaining productivity even in the absence of fertilizer. The highest dose (20 t ha^−1^), in turn, did not favor ear development.

## 3. Discussion

The positive interaction between organic fertilizerand microorganisms in plant growth can be explained by the fact that this input promotes the availability of essential nutrients for microorganisms, regulates the soil’s physicochemical properties, and creates favorable conditions for microbial activity. An example is nitrogen mineralization, which is accelerated as microorganisms enhance the degradation of organic materials, resulting in the release of mineralized nutrients [20]. Eleduma et al. [21] reported results that differ from those of the present study, although the 20 t ha^−1^ dose also produced greater plant height (119.03 cm), it differed significantly from the 10 t ha^−1^ dose (114.69 cm) in maize plants fertilized with cattle manure.

Similarly, [22] evaluated the influence of different *Trichoderma* spp. strains on the initial growth of maize and observed stimulation in plant height compared with the control treatment.

The treatments with *Trichoderma* promoted a significant increase in leaf area, indicating that *Trichoderma* can reduce plant stress under saline conditions by stimulating antioxidant enzyme activity and gene expression, improving root development, and enhancing water and nutrient uptake. In addition, it increases proline levels, modulates phytohormone balance, and assists in salt removal, thereby reducing the detrimental effects of salinity stress [23,24].

The reduction in leaf area under salinity stress is related to an adaptive mechanism by which plants minimize excessive water loss through transpiration. This occurs because elevated salt concentrations decrease the osmotic potential of the soil solution, which restricts water movement into the roots [25]. Kumar et al. [26], evaluating different salt-tolerant strains under pot conditions, observed a reduction in leaf area in plants irrigated with brackish water; however, when the plants were inoculated with *Trichoderma* isolates, they exhibited higher mean leaf area values. Similarly, [27] found that maize plants grown in soil containing *Trichoderma harzianum* showed greater leaf development, even under irrigation with lower-salinity water, compared with non-inoculated plants.

After its application, cattle manure can be decomposed in the rhizosphere, improving soil fertility through the release of plant-available nutrients and consequently stimulating plant growth [28]. The addition of organic inputs to the soil enhances microbial activity, creating a favorable environment for the development of microbial biomass. This process improves nutrient mineralization and availability in the soil [20]. Similar results, in which organic fertilizer with cattle manure increased leaf area in maize plants, were also reported by [29], who combined manure application with an ionic polymer, and by [21], who evaluated different cattle manure doses.

Different results from this study for stem diameter were reported by [30], who found no significant effect for this variable, either when using *Trichoderma* alone or in combination with cattle manure, in maize plants. *Trichoderma* can partially mitigate the negative effects of salinity on stem growth, depending on the combination of salinity level and fertilizer rate. Organic matter acts as a soil conditioner, improving not only fertility but also aggregation, nutrient cycling, and serving as a food source for soil microbiota [31]. *Trichoderma*, in turn, promotes the synthesis of phytohormones that stimulate cell wall expansion and biomass production, resulting in plants with larger stem diameter [32].

Wei et al. [33], studying sweet sorghum (variety Aertuo326) with seed inoculation of *Trichoderma harzianum*, reported a significant effect of the microorganism, with a 32.09% increase in stem diameter compared to the control treatment. Similarly, [34], evaluating the effect of different *T. harzianum* doses on maize, also found an increase in this variable of up to 34% compared to the control.

The results obtained for leaf number in this study are consistent with those observed by [21], reported that increasing the amount of organic input in the soil enhances microbial activity, leading to greater nutrient release and absorption by plant roots, which in turn promotes an increase in leaf number. The same authors observed an increase in the number of maize leaves with cattle manure application up to a dose of 20 t ha^−1^. In contrast, [30] found that the number of leaves was not significantly affected by treatments with *Trichoderma*-based bioinoculants in maize cultivation.

During the first cultivation cycle, irrigation with higher-salinity water (3 dS m^−1^) led to a higher CO_2_ assimilation rate than irrigation with low-salinity water (0.3 dS m^−1^). This response may be associated with compensatory physiological mechanisms commonly observed in plants under moderate salinity stress. Among these mechanisms are the maintenance of the integrity of the photosynthetic apparatus, increased efficiency in CO_2_ fixation, and cellular osmotic adjustment, which enable plants to sustain their metabolic activity even under adverse conditions [35].

Sousa et al. [10], using the same maize cultivar and under environmental conditions similar to those of the present study, observed a similar pattern, with higher CO_2_ assimilation rates under saline irrigation (3.0 dS m^−1^). Conversely, [36] reported no significant differences in CO_2_ assimilation in maize grown under different salinity levels (0.86 and 3.26 dS m^−1^), indicating that physiological responses may vary depending on the cultivar, phenological stage, and edaphoclimatic conditions.

According to [37], under saline conditions, plants can activate signaling responses that trigger the production of protective or regulatory compounds, thereby favoring the maintenance of photosynthetic activity. This was also demonstrated by [38], who reported a positive effect of different *Trichoderma* strains on physiological variables in rice plants.

The positive effect of *Trichoderma* may be attributed to its contribution to improved root distribution and stimulation of leaf area expansion, both of which enhance photosynthetic activity in plants. Consistent with this finding, [39] observed a significant increase in the CO_2_ assimilation rate of maize plants inoculated with *T. asperellum* strain T34.

No significant variation in transpiration was detected among the fertilizer rates within each salinity level. When exposed to saline environments, plants develop mechanisms to survive and maintain homeostasis, one of the fastest responses being stomatal closure, which consequently reduces transpiration [3,40]. However, plants may achieve osmotic adjustment, maintaining turgor and growth if salt concentrations do not reach toxic levels [41], which may explain the results found in the present study. Sousa et al. [10] also reported increased transpiration in maize plants irrigated with water of 3 dS m^−1^, whereas [42] found contrasting results, observing a reduction in transpiration as the electrical conductivity of irrigation water increased.

*Trichoderma* is an endophytic fungus capable of establishing a symbiotic relationship with plant roots, producing compounds such as antioxidant enzymes that enhance plant tolerance to salinity stress [23]. According to [42], *T. harzianum* induces the synthesis of phytohormones such as gibberellins, auxins, and cytokinins, which promote root growth and nutrient uptake, resulting in increased chlorophyll production and photosynthetic efficiency even under saline conditions. These factors may explain the results observed for stomatal conductance in maize plants.

Consistent with this study the findings of [43], they demonstrated that the presence of *Trichoderma asperellum* increased stomatal conductance in two maize cultivars grown under saline conditions. Similarly, [42] reported a positive effect of *Trichoderma harzianum* inoculation on stomatal conductance in both maize and rice plants cultivated under saline hydroponic conditions.

According to [44], inoculation with *Trichoderma harzianum* induces chlorophyll synthesis in plants under various stress conditions. Likewise, [45] reported an increase in the chlorophyll index of wheat plants treated with *Trichoderma reesei*, both in treatments without NaCl addition and at different NaCl concentrations. Kumar et al. [26] also observed that total chlorophyll content increased due to inoculation with *Trichoderma* isolates, with the highest chlorophyll levels recorded in plants treated with TRC3 (*Trichoderma harzianum*) and the lowest in the control treatment.

The increase in ear length with manure application can be attributed to enhanced nutrient availability, favored by faster mineralization under high-temperature conditions, which provides the plant with the essential elements for better ear development. Contrary to the findings of the present study, [46], working with different cattle manure doses on maize productivity, found no significant difference between the 10 and 20 t ha^−1^ doses for the ear length variable.

Similar findings were reported by [32], who tested *Trichoderma harzianum* inoculation in maize plants and observed significant increases in ear diameter when the microorganism was applied.

Sousa et al. [10] reported similar results, in which irrigation with higher electrical conductivity water (3 dS m^−1^) combined with inoculation by *Bacillus aryabhattai* did not affect ear yield with husk in maize plants.

Steffen et al. [32] also observed a higher number of grains per row in corn ears in treatments inoculated with *Trichoderma* compared to the non-inoculated treatment. This higher grain filling can be attributed to the fungus’ ability to stimulate root growth, thereby improving water and nutrient uptake. This is reflected in better production rates.

Santos et al. [47] found no statistical difference in the use of organic fertilisation in NRPE in different maize cultivars. Guerra et al. [48], however, found a significant effect with an increase in NRPE when using poultry litter on maize plants.

Under adverse conditions, *Trichoderma* may redirect its metabolic activity toward survival processes, thereby reducing its plant growth-promoting effects [49]. Thus, unlike what was observed in the first cycle, where the bioinoculant enhanced EYH, its activity may have been limited in the second cycle, negatively impacting ear productivity. Nascimento et al. [15], working with the same *Trichoderma* strain and maize cultivar used in this study, also reported no significant effect on ear yield with husk.

Syamsiyah et al. [50] working with different mineral fertilizer rates combined with *Trichoderma* inoculation, also reported increased ear yield with husk, with the combination of three-quarters of the mineral fertilizer rate plus *Trichoderma* showing the highest mean yield among treatments. Souza et al. [30], investigating the effect of bioinputs on maize production, also reported a positive effect of the combination of *Trichoderma* and organic inputs on ear yield, both with and without husk. A positive effect of *Trichoderma* associated with cattle manure on the increase of ear yield without husk (EYWH) was also reported by [15] in maize plants grown under experimental conditions similar to those of the present study.

The research was conducted under field conditions, which provides greater representativeness of the observed responses of green maize to inoculation with *Trichoderma harzianum* and to different doses of organic fertilisation under saline stress. However, the results reflect the specific soil and climate conditions of the experimental area, which can strongly influence the physiological and productive responses of the crop.

Therefore, future research should seek to validate these results in different soil types, climatic regions, and cropping systems in order to assess the consibstency of the observed responses and their applicability in different production contexts. In addition, complementary studies that include assessments of biochemical and molecular parameters—such as osmolyte accumulation, ionic balance, and antioxidant activity—may contribute to a more comprehensive understanding of the tolerance mechanisms induced by *Trichoderma harzianum* in plants under saline stress.

## 4. Materials and Methods

### 4.1. Study Area

The experiment was conducted from August to November during the 2022 and 2023 growing seasons at the Piróas Experimental Farm, belonging to the University for International Integration of the Afro-Brazilian Lusophony (UNILAB). The site is located in the district of Barra Nova, municipality of Redenção, Ceará, Brazil, at geographic coordinates 04°14′53″ S, 38°45′10″ W, and an average altitude of 240 m.

The climate of the region is classified as BSh’, i.e., hot semiarid tropical with humid tendencies, characterized by very high temperatures and rainfall concentrated mainly during the summer and autumn seasons [51]. During the experimental period, total precipitation was 9.25 mm in 2022 and 24.12 mm in 2023. The mean temperature was 29.57 °C in 2022 and 29.82 °C in 2023 (Figure 12).

Soil samples were collected before the installation of the experiments at a depth of 0–20 cm and sent to the Soil and Water Laboratory of the Department of Soil Science, Federal University of Ceará (UFC), for the determination of chemical attributes (Table 4), following the methodology described in the Manual of Soil Analysis Methods by Embrapa [52]. The soil of the experimental area has classified as a Argisol with a sandy loam texture [53].

According to Table 4, the soil showed moderate fertility, with pH values ranging from slightly acidic to near neutral (5.6–6.4) and increased organic matter and nutrient content in 2023 compared to 2022. There was also an increase in exchangeable calcium and sodium levels and a reduction in exchangeable magnesium and aluminum. In addition, there was a reduction in ECe in the 2023 cycle compared to 2022.

### 4.2. Experimental Design

The experiment was arranged in a randomized complete block design in a split–split plot scheme with four replications. The main plots consisted of two levels of irrigation water electrical conductivity (ECw: A1—0.3 dS m^−1^ and A2—3.0 dS m^−1^). The subplots corresponded to three levels of organic fertilizer (OF), using well-composted cattle manure as the nutrient source (OF: 0, 10, and 20 t ha^−1^). The sub-subplots comprised soil inoculation (TI) and non-inoculation (NI) with the *Trichoderma harzianum*-based biostimulant (strain ESALQ 1306). The treatment diagram is available in Figure 13.

### 4.3. Irrigation Management

The irrigation system was a drip irrigation system with emitters delivering a flow rate of 8 L h^−1^. The distribution uniformity coefficient, evaluated according to the methodology of [54], was 92%. Irrigation management was based on the reference evapotranspiration (ETo), calculated every two days using data from a Class A evaporation pan installed near the experimental area. The following crop coefficients (Kc) were adopted: 0.86 (up to 40 days after sowing—DAS); 1.23 (from 41 to 53 DAS); 0.97 (from 54 to 73 DAS); and 0.52 (from 74 DAS until the end of the cycle) [55].

The potential crop evapotranspiration (ETc) was determined according to [56], as follows:ETc = ECA × Kp × Kc(1)
whereETc—crop evapotranspiration, in mm day^−1^;ECA—evaporation measured in the class A pan, in mm/day^−1^;Kp—class A pan coefficient, dimensionless;Kc—crop coefficient, dimensionless.

The irrigation time was calculated using Equation:(2)$$It=\;\frac{ETc\;\times \;Sd\;\times \;Sl}{Af\;\times \;q}\;\times \;60$$
whereIt—irrigation time (min);ETc—crop evapotranspiration for the period (mm);Sd—spacing between emitters (m);Sl—lateral spacing (m);Af—application efficiency (0.92);q—flow rate (L h^−1^).

The irrigation waters were prepared as follows: the low-salinity water was obtained from a local supply source (a reservoir located on the experimental farm) with an electrical conductivity of 0.3 dS m^−1^, while the high-salinity water (3 dS m^−1^) was prepared by adding sodium chloride (NaCl), calcium chloride (CaCl_2_ 2H_2_O), and magnesium chloride (MgCl_2_ 6H_2_O) in a 7:2:1 ratio, as recommended by [57]. This proportion reflects the predominant ionic composition of saline well water in the semiarid region. The electrical conductivity of the irrigation water was monitored using a portable conductivity meter (Manufactured by HM Digital, country of origin: Designed in Korea and manufactured in China).

In the first year of cultivation, the volume applied was 500 mm of irrigation. In the second year of cultivation, it was 542 mm of irrigation. Water samples were collected and sent to the laboratory for chemical characterization (Table 5), following the methodology proposed by [52].

### 4.4. Experiment Conduction

The experiment was carried out under field conditions. Twenty days before sowing (DAS), the experimental area was prepared and cleared. A basal fertilizer was then applied according to the recommendation of [59] for irrigated maize. Fifteen planting holes were opened per plot, and five maize seeds were sown in each hole. Before the beginning of the second growing cycle, liming was performed according to the recommendation of [60], aiming to correct the soil pH from 5.6 to 6.4, which is considered within the optimal range for maize cultivation.

The maize cultivar used was Catingueiro, with a spacing of 0.8 m × 0.2 m between rows and between plants, respectively. This cultivar was chosen because it is the most widely grown by farmers in the study region and is well adapted to the local climatic conditions. Thinning was performed ten days after sowing, leaving one plant per hole.

### 4.5. Inoculant and Fertilizer Strategy

The organic fertilizer used was well-composted cattle manure, applied in lateral furrows near the planting rows and incorporated into ridges measuring 0.20 m in width and 0.15 m in height from the soil surface. The application was carried out in two stages (at planting and as topdressing), according to the soil chemical analysis (Table 1) and the recommendation proposed by for green maize (100% = 20 t ha^−1^). The manure doses were 0, 10, and 20 t ha^−1^, corresponding to 0%, 50%, and 100% of the recommended rate. The chemical characteristics of the cattle manure were determined following the methodology proposed by [52] are presented in Table 6.

The biostimulant used in the experiment was *Trichoderma harzianum*, a commercial product (*Trichodermil*, strain ESALQ 1306), in the form of a concentrated suspension containing 2.0 × 10^9^ viable conidia mL^−1^. The product belongs to the Koppert brand, Piracicaba - SP, Brazil. The application of *Trichoderma* followed the manufacturer’s recommendation of 4 L ha^−1^ (adjusted according to the plot area) and was applied to the soil near the planting furrows, always in the late afternoon, divided into four applications throughout the vegetative growth stage, with the first application on the day of sowing and the others every 15 days to ensure inoculation efficiency.

### 4.6. Analysed Atributes

#### 4.6.1. Plant Growth

At 42 days after sowing (DAS), the following variables were evaluated: plant height (PH, cm), measured using a graduated measuring tape; number of leaves (NL), determined by direct counting; stem diameter (SD, mm), measured with a digital caliper; and leaf area (LA, cm^2^), calculated as A = C × L × 0.75, where C is the leaf length, L is the leaf width, and 0.75 is the correction coefficient [61].

#### 4.6.2. Gas Exchange and SPAD Index

49 days after sowing (DAS), gas exchange measurements were taken using the third fully expanded leaf from the plant apex as the reference. The CO_2_ assimilation rate (A), transpiration rate (E), and stomatal conductance (gs) were measured with an infrared gas analyzer (IRGA, model LC-Pro-SD, ADC Bioscientific Ltd., Hoddesdon, Hertfordshire, UK) under the following conditions: ambient air temperature, CO_2_ concentration of 400 ppm, and photosynthetically active radiation (PAR) of 1800 µmol m^−2^ s^−1^. Readings were performed between 9:00 and 11:00 a.m. The relative chlorophyll content (RCI, SPAD) was determined on the same leaves using a portable chlorophyll meter (SPAD-502 Plus, Minolta, Tokyo, Japan).

#### 4.6.3. Crop Yield

80 days after sowing (DAS), when the maize reached the R4 (stage according to the scale of Ritchie et al. 1993 [62]) stage corresponding to the dough/milky grain phase, with approximately 70% grain moisture, five ears were collected from the useful plot area, and the following variables were evaluated: ear diameter (ED, mm), measured with a digital caliper; ear length without husk (EL, cm), measured with a graduated ruler; yield of ears with husk (YEH, kg ha^−1^); and yield of ears without husk (YEWH, kg ha^−1^), estimated from the average ear weight and the plant population density established per hectare (62,500 plants ha^−1^).

The observed data were subjected to the Kolmogorov–Smirnov normality test, followed by analysis of variance (ANOVA). When significant differences were detected by the F-test, the means were compared using Tukey’s test at a 5% significance level (*). Statistical analyses were performed using the ASSISTAT 7.7 Beta software [63].

## 5. Conclusions

Irrigation with brackish water affected maize plant growth, leaf gas exchange, and productivity in both cultivation cycles; however, these effects were less pronounced in the presence of *Trichoderma harzianum* and with organic fertilizer using well-composted cattle manure. Nonetheless, inoculation with *Trichoderma harzianum* proved effective in improving the agronomic performance of maize plants, particularly under lower irrigation water electrical conductivity conditions. Moreover, organic fertilizer at a rate of 10 t ha^−1^ combined with *Trichoderma harzianum* inoculation resulted in greater production efficiency, standing out as a promising strategy for yield maximization.

It is important to validate the findings across different soil types, climatic regions, and cropping systems to confirm the consistency of the responses observed in this study. Complementary research, including biochemical and molecular assessments, may further advance our understanding of the mechanisms of salt stress tolerance induced by *Trichoderma harzianum.*

## Figures and Tables

**Figure 1 plants-14-03643-f001:**
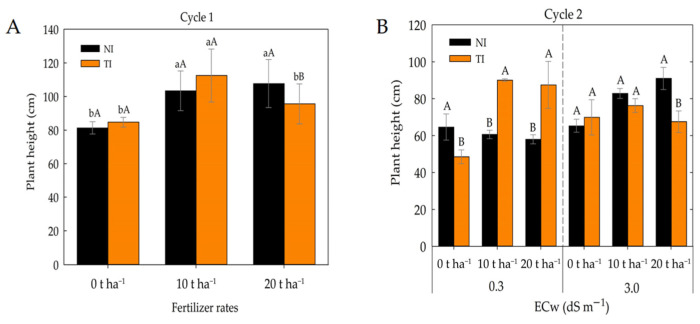
Plant height of maize subjected to different fertilizer rates in non-inoculated (NI) and *Trichoderma*-inoculated (TI) soil during the first cycle (**A**). Plant height of maize under different levels of irrigation water electrical conductivity (ECw) and fertilizer rates in non-inoculated (NI) and *Trichoderma*-inoculated (TI) soil during the second cycle (**B**). Lowercase letters compare mean values among fertilizer rates within non-inoculated (NI) and *Trichoderma*-inoculated (TI) soil; uppercase letters compare means between non-inoculated (NI) and *Trichoderma*-inoculated (TI) soil within each fertilizer rates according to Tukey’s test (*p* ≤ 0.05). Uppercase letters also compare means between plants grown in non-inoculated (NI) and *Trichoderma*-inoculated (TI) soil under the same electrical conductivity and fertilizer rates according to Tukey’s test (*p* ≤ 0.05).

**Figure 2 plants-14-03643-f002:**
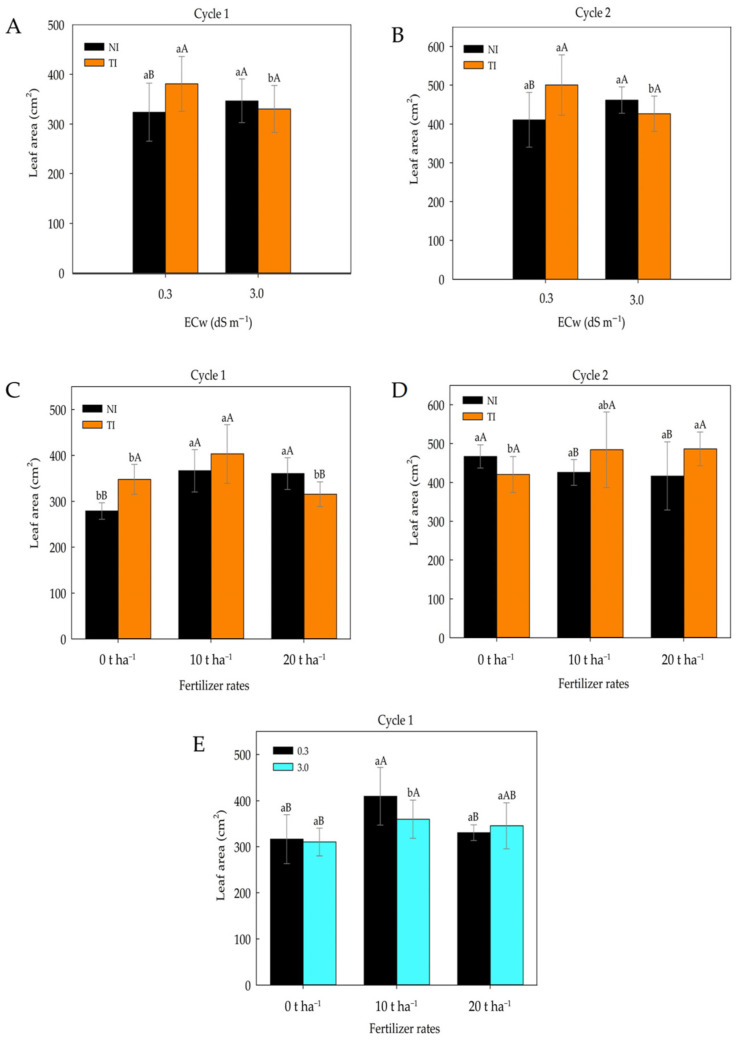
Leaf area of maize plants subjected to different levels of irrigation water electrical conductivity (ECw) in non-inoculated (NI) and *Trichoderma*-inoculated (TI) soil during cultivation cycles 1 (**A**) and 2 (**B**). Leaf area of maize plants subjected to different fertilizer rates in non-inoculated (NI) and *Trichoderma*-inoculated (TI) soil during cultivation cycles 1 (**C**) and 2 (**D**). Leaf area of maize plants subjected to different fertilizer rates and irrigation water electrical conductivity levels (ECw) (**E**). Lowercase letters compare means between non-inoculated (NI) and *Trichoderma*-inoculated (TI) soil for each ECw level; uppercase letters compare means of NI and TI within each ECw level according to Tukey’s test (*p* ≤ 0.05). Lowercase letters also compare mean values among fertilizer rates within non-inoculated (NI) and *Trichoderma*-inoculated (TI) soil, while uppercase letters compare means between NI and TI within each fertilizer rates according to Tukey’s test (*p* ≤ 0.05). Lowercase letters compare means of ECw levels within each fertilizer rate, and uppercase letters compare means among fertilizer rates within the same ECw level according to Tukey’s test (*p* ≤ 0.05).

**Figure 3 plants-14-03643-f003:**
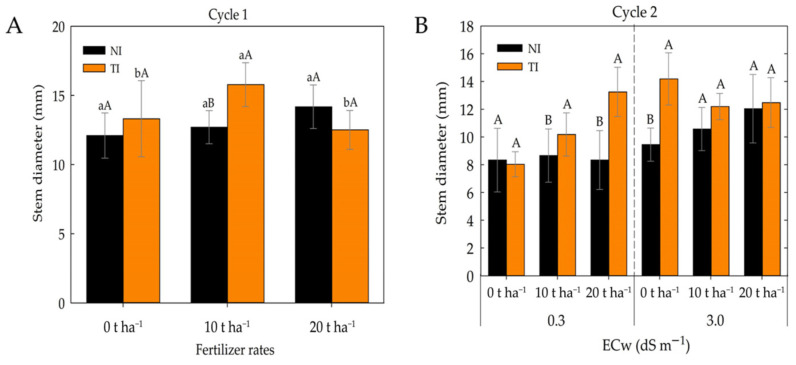
Stem diameter of maize plants under different fertilizer rates and irrigation water electrical conductivity (ECw) levels in cultivation cycle 1 (**A**) and under different ECw levels and fertilizer rates in non-inoculated (NI) and *Trichoderma harzianum*-inoculated (TI) soil in cultivation cycle 2 (**B**). Lowercase letters compare mean values among fertilizer rates within non-inoculated (NI) and *Trichoderma*-inoculated (TI) soil; uppercase letters compare means between NI and TI within each fertilizer rate according to Tukey’s test (*p* ≤ 0.05). Uppercase letters also compare means between plants grown in NI and TI soil under the same electrical conductivity and fertilizer rates according to Tukey’s test (*p* ≤ 0.05).

**Figure 4 plants-14-03643-f004:**
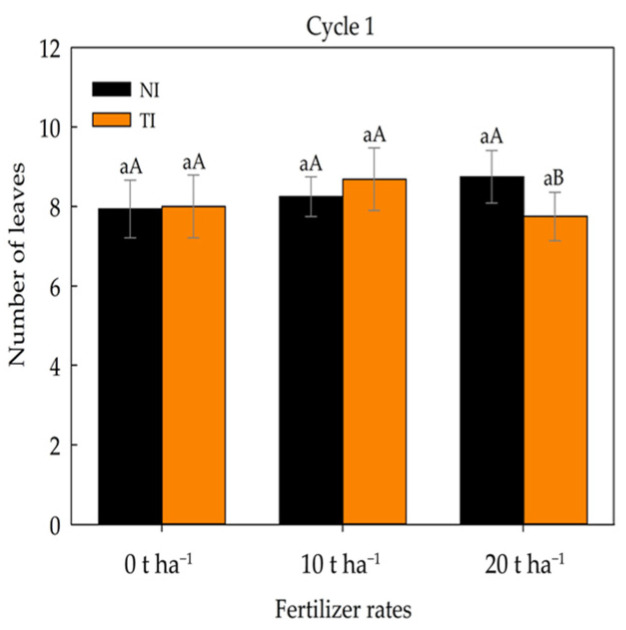
Number of leaves in maize plants under different fertilizer rates in non-inoculated (NI) and *Trichoderma*-inoculated (TI) soil. A: Lowercase letters compare mean values among fertilizer rates within non-inoculated (NI) and *Trichoderma*-inoculated (TI) soil; uppercase letters compare means between NI and TI within each fertilizer rate according to Tukey’s test (*p* ≤ 0.05).

**Figure 5 plants-14-03643-f005:**
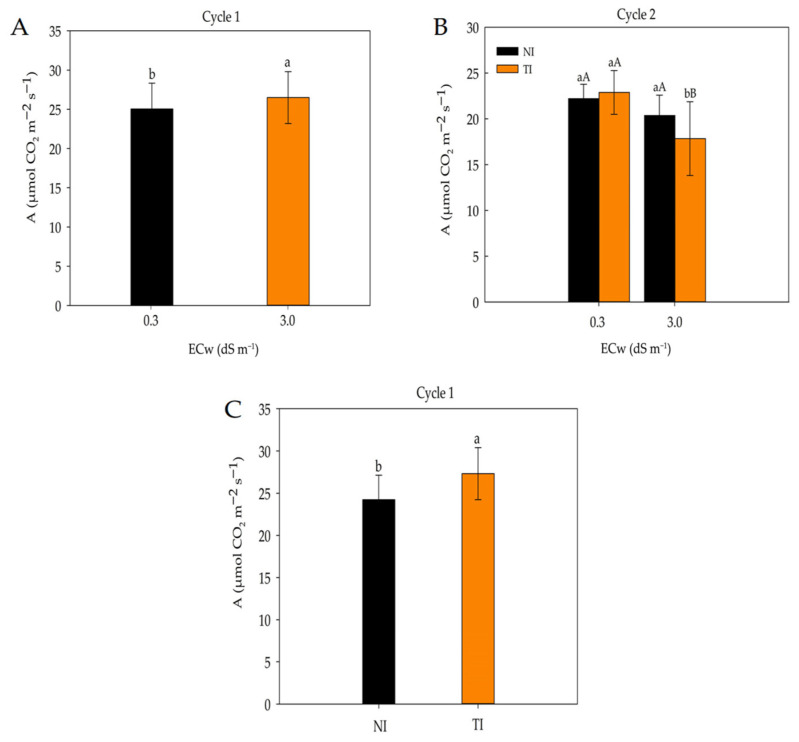
CO_2_ assimilation rate in maize plants subjected to different levels of irrigation water electrical conductivity (ECw), cycle 1 (**A**). CO_2_ assimilation rate in maize plants subjected to irrigation water electrical conductivity (ECw) in non-inoculated (NI) and *Trichoderma*-inoculated (TI) soil, cycle 2 (**B**). CO_2_ assimilation rate in maize plants grown in non-inoculated (NI) and *Trichoderma*-inoculated (TI) soil, cycle 1 (**C**). Identical letters do not differ by Tukey’s test (*p* ≤ 0.05). Lowercase letters compare means between non-inoculated (NI) and *Trichoderma*-inoculated (TI) soil for each ECw level; uppercase letters compare means of NI and TI within each ECw level according to Tukey’s test (*p* ≤ 0.05).

**Figure 6 plants-14-03643-f006:**
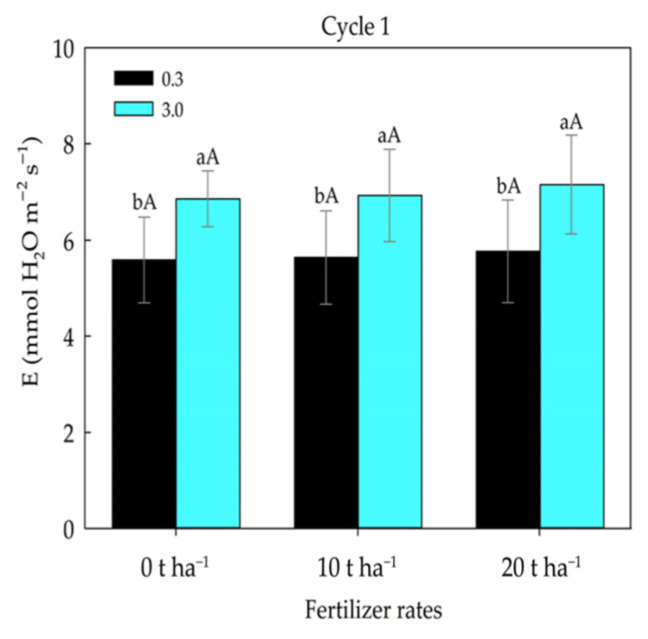
Transpiration in maize plants subjected to different fertilizer rates and irrigation water electrical conductivity (ECw). Lowercase letters compare means of ECw levels within each fertilizer rate, and uppercase letters compare means among fertilizer rates within the same ECw level according to Tukey’s test (*p* ≤ 0.05).

**Figure 7 plants-14-03643-f007:**
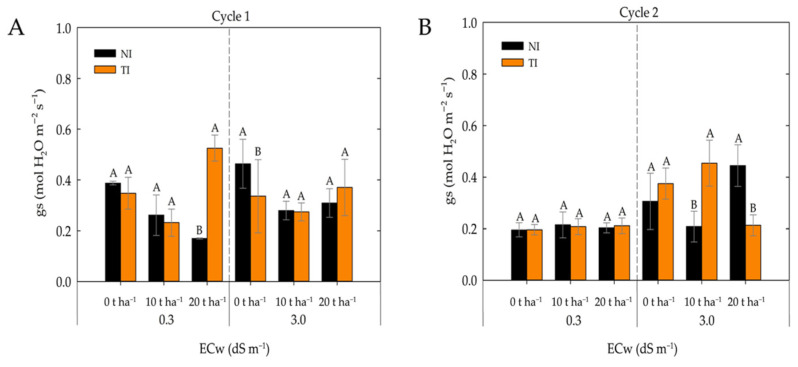
Stomatal conductance in maize plants under different levels of irrigation water electrical conductivity (ECw) and fertilizer rates in non-inoculated (NI) and *Trichoderma*-inoculated (TI) soil during cultivation cycle 1 (**A**) and cycle 2 (**B**). A and B: Uppercase letters compare means between plants grown in non-inoculated (NI) and *Trichoderma*-inoculated (TI) soil under the same electrical conductivity and fertilizer rate according to Tukey’s test (*p* ≤ 0.05).

**Figure 8 plants-14-03643-f008:**
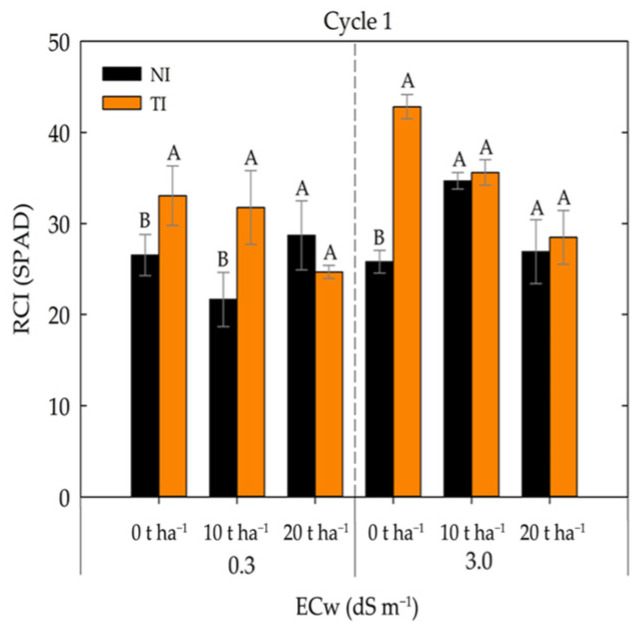
Chlorophyll index in maize plants under different levels of irrigation water electrical conductivity and fertilizer rates in non-inoculated (NI) and *Trichoderma*-inoculated (TI) soil. Uppercase letters compare means between plants grown in non-inoculated (NI) and *Trichoderma*-inoculated (TI) soil under the same electrical conductivity and fertilizer rate according to Tukey’s test (*p* ≤ 0.05).

**Figure 9 plants-14-03643-f009:**
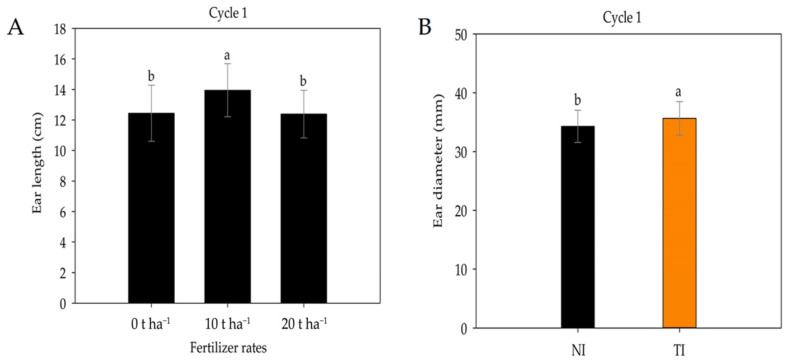
Ear length under different fertilizer rates (**A**) and ear diameter in non-inoculated (NI) and *Trichoderma*-inoculated (TI) soil (**B**) during cultivation cycle 1. Identical letters do not differ by Tukey’s test (*p* ≤ 0.05).

**Figure 10 plants-14-03643-f010:**
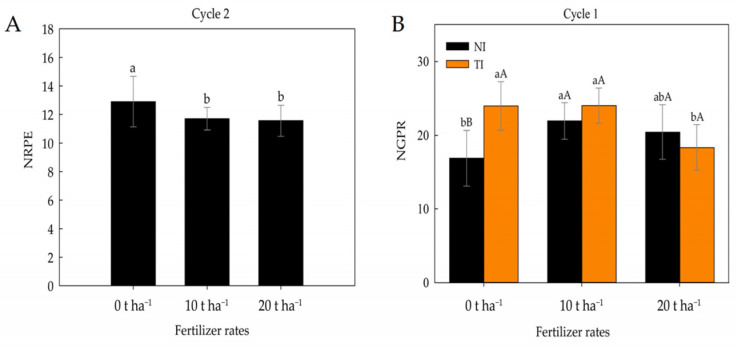
Number of rows per ear (NRPE) under different fertilizer rates in cycle 2 (**A**) and number of grains per row (NGPR) under different fertilizer rates in non-inoculated (NI) and *Trichoderma*-inoculated (IT) soil in cycle 1 (**B**). Identical letters do not differ by Tukey’s test (*p* ≤ 0.05). Lowercase letters compare mean values among fertilizer rates within non-inoculated (NI) and *Trichoderma*-inoculated (TI) soil, and uppercase letters compare means between NI and TI within each fertilizer rate according to Tukey’s test (*p* ≤ 0.05).

**Figure 11 plants-14-03643-f011:**
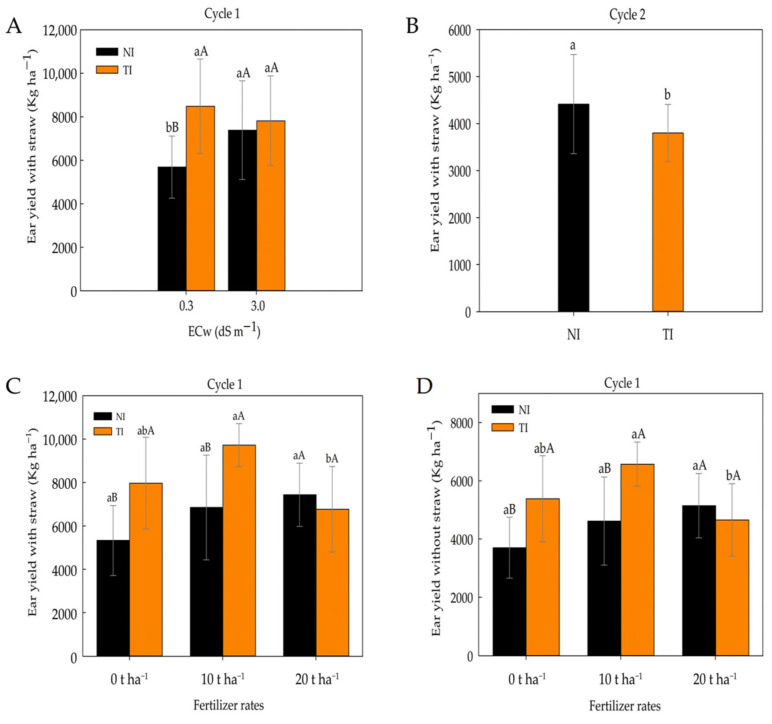
Ear yield with husk (EYH) in non-inoculated (NI) and *Trichoderma*-inoculated (TI) soil under different irrigation water electrical conductivity (ECw) levels. Lowercase letters compare NI and TI in cycle 1 (**A**). Ear yield with husk (EYH) in non-inoculated (NI) and *Trichoderma*-inoculated (TI) soil in cycle 2 (**B**). Ear yield with husk (EYH) under different fertilizer rates in non-inoculated (NI) and *Trichoderma*-inoculated (TI) soil in cycle 1 (**C**). Ear yield without husk (EYWH) under different fertilizer rates in non-inoculated (NI) and *Trichoderma*-inoculated (IT) soil in cycle 1 (**D**). Lowercase letters compare means of ECw levels within non-inoculated (NI) and *Trichoderma*-inoculated (TI) soil, while uppercase letters compare means of NI and TI within each ECw level according to Tukey’s test (*p* ≤ 0.05). Identical letters do not differ by Tukey’s test (*p* ≤ 0.05). Lowercase letters compare mean values among fertilizer rates within non-inoculated (NI) and *Trichoderma*-inoculated (TI) soil, and uppercase letters compare means between NI and TI within each fertilizer rate according to Tukey’s test (*p* ≤ 0.05).

**Figure 12 plants-14-03643-f012:**
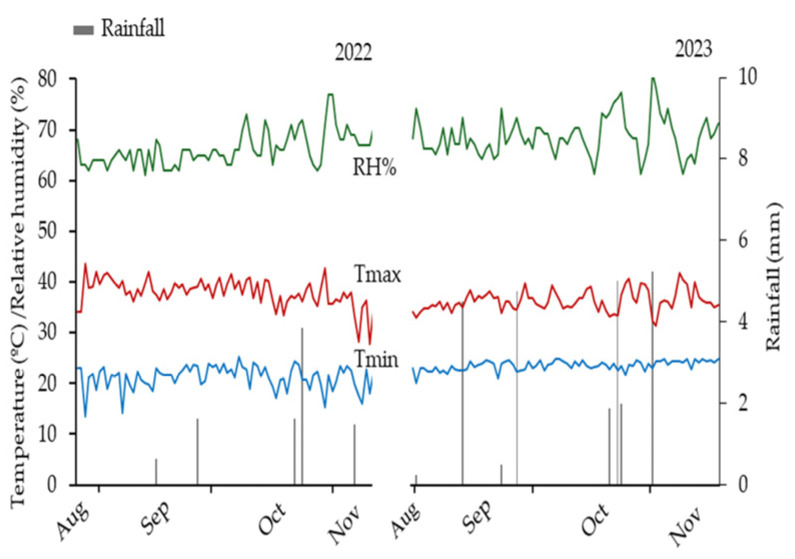
Temperature (°C), relative humidity (%), and rainfall (mm) recorded at the Piróas Experimental Farm during the maize growing months in 2022 and 2023. Tmin: minimum temperature; Tmax: maximum temperature; RH: relative air humidity.

**Figure 13 plants-14-03643-f013:**
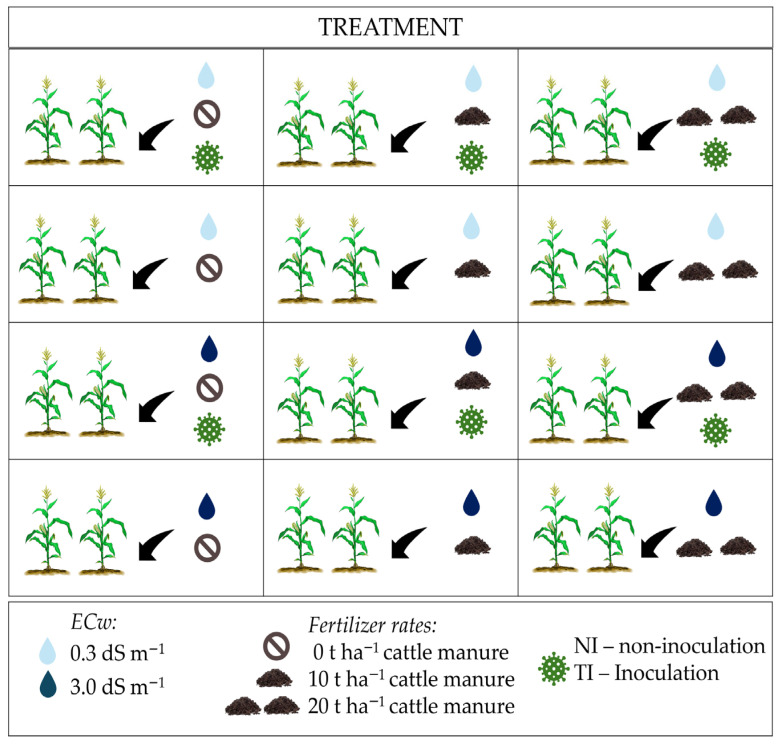
Diagram of the experimental design showing the composition and interaction of the study factors—electrical conductivity of water, fertiliser doses and inoculation.

**Table 1 plants-14-03643-t001:** Summary of the analysis of variance (mean square) for plant height (PH), leaf area (LA), stem diameter (SD), and number of leaves (NL) in maize plants subjected to different salinity levels in water irrigation and organic fertilizer rates, with and without *Trichoderma* inoculation, during two cultivation cycles.

Source of Variation	DF	Mean							
		PH	LA	SD	NL	PH	LA	SD	NL
		Cycle 1				Cycle 2			
Blocks	3	67.26 ^ns^	653.41 ^ns^	1.79 ^ns^	0.57 ^ns^	22.58 ^ns^	1906.40 ^ns^	7.74 ^ns^	1.13 ^ns^
Water electrical conductivity (ECw)	1	1710.04 *	2271.17 ^ns^	0.35 ^ns^	0.33 ^ns^	630.75 *	1566.66 ^ns^	66.43 *	2.75 ^ns^
Error	3	61.21	471.97	5.03	0.47	42.36	4651.07	6.010	0.53
Fertilizer rates (FR)	2	2691.70 **	21,029.08 **	9.58 ^ns^	1.00 ^ns^	1150.40 **	544.33 ^ns^	9.97 *	0.14 ^ns^
Error	12	147.72	1022.33	5.00	0.80	30.35	2740.89	2.20	0.52
Inoculation (I)	1	0.25 ^ns^	4939.16 ^ns^	9.12 ^ns^	0.33 ^ns^	98.61 ^ns^	8926.93 ^ns^	55.66 **	0.04 ^ns^
Error	18	80.79	1792.39	2.63	0.39	64.91	2876.39	4.54	0.88
ECw x FR	2	365.10 ^ns^	4374.69 *	8.11 ^ns^	1.03 ^ns^	48.26 ^ns^	10,645.36 ^ns^	5.15 ^ns^	0.09 ^ns^
ECw x I	1	3.79	16,061.31 **	1.74 ^ns^	0.02 ^ns^	1552.68 **	46,626.97 **	0.14 ^ns^	0.63 ^ns^
FE x I	2	480.19 *	13,735.02 **	22.81 **	2.22 *	295.03 *	16,325.54 *	1.22 ^ns^	0.48 ^ns^
ECw x FR x I	2	13.32 ^ns^	426.72 ^ns^	1.37 ^ns^	0.19 ^ns^	1487.18 **	5575.24 ^ns^	22.63 *	0.13 ^ns^
CV (%)-ECw		8.02	6.29	16.7	8.35	9.06	15.16	23.04	8.37
CV (%)-FR		12.46	9.26	16.67	10.91	7.67	11.63	13.97	8.35
CV (%)-I		9.22	12.26	12.09	7.61	11.22	11.92	20.02	10.8

SV—Sources of variation; DF—Degrees of freedom; CV—Coefficient of variation; ** Significant at the 1% probability level (*p* < 0.01); * Significant at the 5% probability level (0.01 ≤ *p* < 0.05); ns—Not significant (*p* ≥ 0.05); PH—plant height; LA—leaf area; SD—stem diameter; NL—number of leaves; ECw water electrical conductivity;—FR—fertilizer rates; I—inoculation.

**Table 2 plants-14-03643-t002:** Summary of the analysis of variance (mean square) for CO_2_ assimilation rate (A), transpiration rate (E), stomatal conductance (gs), and chlorophyll index (SPAD) in maize plants subjected to different salinity levels in irrigation water and organic fertilizer rates, with and without *Trichoderma* inoculation, during two cultivation cycles.

Source of Variation	DF	Mean							
		A	E	gs	SPAD	A	E	gs	SPAD
		Cycle 1				Cycle 2			
Blocks	3	4.40 ^ns^	5.76 **	0.45 ^ns^	6.23 ^ns^	9.56 ^ns^	1.00 ^ns^	0.36 ^ns^	10.23 ^ns^
Water electrical conductivity (ECw)	1	25.30 *	20.81 **	0.05 ^ns^	259.93 **	142.07 ^ns^	0.23 ^ns^	36.16 *	39.42 ^ns^
Error	3	1.76	0.17	0.15	4.170	20.89	1.35	1.07	37.42
Fertilizer rates (FR)	2	15.54 ^ns^	0.24 ^ns^	0.86 **	103.58 **	7.34 ^ns^	0.01 ^ns^	0.00 *	6.44 ^ns^
Error	12	4.75	0.75	0.05	5.29	10.22	0.39	1.03	11.83
Inoculation (I)	1	114.54 **	2.28 *	0.21 ^ns^	344.00 **	10.36 ^ns^	0.36 ^ns^	0.43 ^ns^	2.38 ^ns^
Error	18	10.080	0.50	0.06	13.23	5.20	0.18	0.72	8.88
ECw x FR	2	16.05 ^ns^	0.01 *	0.029 ^ns^	55.36 **	6.33 ^ns^	0.10 ^ns^	0.13 ^ns^	4.57 ^ns^
ECw x I	1	1.53 ^ns^	0.14 ^ns^	0.59 **	15.98 ^ns^	31.04 *	0.10 ^ns^	0.39 ^ns^	2.85 ^ns^
F x I	2	8.32 ^ns^	0.25 ^ns^	1.31 **	169.07 **	6.15 ^ns^	0.09 ^ns^	9.95 **	0.10 ^ns^
ECw x FR x I	2	33.88 ^ns^	1.27 ^ns^	1.31 *	104.90 **	2.67 ^ns^	0.05 ^ns^	11.27 **	16.73 ^ns^
CV (%)-ECw		5.16	6.6	38.02	6.8	21.94	33.07	38.73	24.47
CV (%)-FR		8.46	13.73	21.93	7.66	15.35	17.75	29.67	13.76
CV (%)-I		12.32	11.21	24.75	12.1	10.9	12.14	27.69	11.92

SV: sources of variation; DF: degrees of freedom; * significant by the F-test at 5%; ** significant by the F-test at 1%; ns: not significant; CV: coefficient of variation; A—CO_2_ assimilation rate; E—transpiration rate; gs—stomatal conductance; SPAD—chlorophyll index; ECw—water electrical conductivity;—FR—fertilizer rates; I—inoculation.

**Table 3 plants-14-03643-t003:** Summary of the analysis of variance for ear diameter (ED), ear length (EL), number of rows per ear (NRPE), number of grains per row (NGPR), ear yield without husk (EYWH), and ear yield with husk (EYH) in maize plants subjected to different salinity levels in irrigation water and organic fertilizer rates, with and without *Trichoderma* inoculation, during two cultivation cycles.

Cycle 1							
Source of Variation	DF	Mean					
		ED	EL	NRPE	NGPR	EYWH	EYH
Blocks	3	0.81 ^ns^	1.51 ^ns^	1.29 ^ns^	8.99 ^ns^	474,952.30 ^ns^	1,622,053.66 ^ns^
Water electrical conductivity (ECw)	1	50.98 ^ns^	0.005 ^ns^	3.03 ^ns^	40.60 ^ns^	2,115,040.98 ^ns^	3,170,344.29 ^ns^
Error	3	7.05	4.93	0.89	23.01	1,351,729.72	2,370,675.00
Fertilizer (F)	2	17.05 ^ns^	12.64 *	1.64 ^ns^	54.67 *	4,570,054.97 ^ns^	11,474,622.9 ^ns^
Error	12	10.5	2.57	0.98	9.01	2,101,188.29	4,247,645.59
Inoculation (I)	1	24.65 *	3.40 ^ns^	0.08 ^ns^	66.58 *	13,151,278.20 **	31,328,848.00 **
Error	18	5.410	3.23	0.51	11.86	1,546,104.74	3,599,564.47
ECw x F	2	0.86 ^ns^	2.12 ^ns^	0.01 ^ns^	0.82 ^ns^	943,167.51 ^ns^	2,030,409.7 ^ns^
ECw x I	1	9.78 ^ns^	5.29 ^ns^	0.45 ^ns^	9.39 ^ns^	6,260,221.05 ^ns^	16,901,545.17 *
F x I	2	13.94 ^ns^	8.93 ^ns^	0.12 ^ns^	84.64 **	7,166,876.59 *	15,687,688.93 *
ECw x F x I	2	1.26 ^ns^	1.20 ^ns^	0.98 ^ns^	5.93 ^ns^	1,356,593.65 ^ns^	3,846,194.34 ^ns^
CV (%)-ECw		7.61	17.19	7.81	22.91	23.19	20.96
CV (%)-F		9.28	12.4	8.22	14.34	28.91	28.05
CV (%)-I		6.66	13.9	5.91	16.45	24.8	25.83
Cycle 2							
Source of Variation	DF	Mean					
		ED	EL	NRPE	NGPR	EYWH	EYH
Blocks	3	5.73 ^ns^	8.43 ^ns^	0.84 ^ns^	3.76 ^ns^	607,497.91 ^ns^	298,367.42 ^ns^
Water electrical conductivity (ECw)	1	0.00 ^ns^	4.74 ^ns^	0.001 ^ns^	0.50 ^ns^	511,404.72 ^ns^	2,656,205.87 ^ns^
Error	3	0.78	4.32	0.3	10.63	283,766.43	454,567.73
Fertilizer (F)	2	3.25 ^ns^	0.52 ^ns^	8.64 *	1.20 ^ns^	459,253.10 ^ns^	356,415.74 ^ns^
Error	12	2.18	2.29	1.56	3.86	276,397.56	709,774.58
Inoculation (I)	1	1.70 ^ns^	6.02 ^ns^	3.91 ^ns^	2.50 ^ns^	1,290,267.48 ^ns^	4,516,546.50 *
Error	18	5.73	1.91	1.92	4.97	597,237.75	798,665.38
ECw x F	2	3.84 ^ns^	1.28 ^ns^	2.66 ^ns^	0.82 ^ns^	205,430.39 ^ns^	243,914.55 ^ns^
ECw x I	1	0.46 ^ns^	1.24 ^ns^	3.39 ^ns^	0.001 ^ns^	2942.85 ^ns^	114,488.18 ^ns^
F x I	2	5.11 ^ns^	0.09 ^ns^	2.33 ^ns^	1.00 ^ns^	573,823.21 ^ns^	1,026,472.65 ^ns^
ECw x F x I	2	2.080 ^ns^	1.51 ^ns^	2.40 ^ns^	4.54 ^ns^	1,169,818.44 ^ns^	2,140,985.88 ^ns^
CV (%)-ECw		2.75	20.24	4.59	23.36	18.09	16.41
CV (%)-F		4.58	14.74	10.37	14.08	17.86	20.51
CV (%)-I		7.43	13.48	11.5	15.99	26.25	21.76

SV: sources of variation; DF: degrees of freedom; * significant by the F-test at 5%; ** significant by the F-test at 1%; ns: not significant; CV: coefficient of variation; ECw—water electrical conductivity; – FR—fertilizer rates; I—inoculation; ED—ear diameter; EL—ear length; EYWH—ear yield without husk; EYH—ear yield with husk.

**Table 4 plants-14-03643-t004:** Chemical characterization of the soil in the experimental area during the two experimental cycles (2022 and 2023).

Attributes	Seasons	
	2022	2023
ECe (dS m^−1^)	0.76	0.56
pH (H_2_O)	5.6	6.4
OM (g kg^−1^)	11.59	16.07
N (g kg^−1^)	0.71	0.99
P (mg dm^−3^)	20	21
K^+^ (cmol_c_ dm^−3^)	0.17	0.19
Na^+^ (cmol_c_ dm^−3^)	0.07	0.12
Ca^2+^ (cmol_c_ dm^−3^)	3.2	5.9
Mg^2+^ (cmol_c_ dm^−3^)	2.6	0.7
H^+^ + Al^3+^ (cmol_c_ dm^−3^)	2.15	1.82
Al^3+^ (cmol_c_ dm^−3^)	0.35	0.05
SB (cmol_c_ dm^−3^)	6.04	6.9
CEC (cmol_c_ dm^−3^)	8.19	8.72
V (%)	74	79
m (%)	5	1
ESP (%)	0.85	1.38
Sand (%)	507	358
Silt (%)	133	189
Clay (%)	77	132

N: nitrogen; P: phosphorus; K^+^: potassium; Na^+^: sodium; Ca^2+^: calcium; Mg^2+^: magnesium; H^+^ + Al^3+^: potential acidity; Al^3+^: exchangeable aluminum; OM: Organic matter; SB: Sum of bases; CEC: Cation exchange capacity; ESP: Exchangeable sodium percentage; ECe: Electrical conductivity of the soil saturation extract; m: Aluminum saturation (Al^3+^); V: Base saturation.

**Table 5 plants-14-03643-t005:** Chemical characterization and classification of the irrigation water used in the experiment according to [58].

Ca^2+^	Mg^2+^	K^+^	Na^+^	Cl^−^	HCO_3-_	pH	ECw	SAR	Class
(mmol_c_ L^−1^)				(mmol_c_ L^−1^)		^_^	(dS m^−1^)	^_^	^_^
8.1	2.5	0.1	25.8	33.8	2.4	6.5	3.0	11.21	C_4_S_3_
Ca^2+^	Mg^2+^	K^+^	Na^+^	Cl^-^	HCO_3-_	pH	ECw	SAR	Class
(mmol_c_ L^−1^)				(mmol_c_ L^−1^)		^_^	(dS m^−1^)	^_^	^_^
1.2	1.6	0.1	0.7	3.1	0.4	7.2	0.34	0.42	C_2_S_1_

Ca^2+^: calcium; Mg^2+^: magnesium; K^+^: potassium; Cl^-^: chloride; HCO_3-_: bicarbonate ECw: water electrical conductivity; SAR: sodium adsorption ratio.

**Table 6 plants-14-03643-t006:** Chemical characteristics of the well-composted cattle manure used in the treatments.

Cattle Manure							
N	P	K	Ca	Mg	Fe	Cu	Zn
-----------------------------------------------------g·kg^−1^-------------------------------------------------							
19.60	4.95	0.67	1.38	3.85	0.55	0.04	0.10

N: nitrogen; P: phosphorus; K: potassium; Ca: calcium; Mg: magnesium; Fe: iron; Cu: copper; Zn: zinc.

## Data Availability

The original contributions presented in this study are included in the article. Further inquiries can be directed to the corresponding authors.

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
