# Peer review of "Cattle Manure Fertilizer and Biostimulant Trichoderma Application to Mitigate Salinity Stress in Green Maize Under an Agroecological System in the Brazilian Semiarid Region"

_plants, 2025, doi:10.3390/plants14233643_

Round 1

Reviewer 1 Report

Comments and Suggestions for Authors

The main question addressed by the research titled ‘Organic fertilizer and bio-stimulant application to mitigate salinity stress in green maize under an agroecological system in the Brazilian semiarid region’  is: What is the ultimate role of  organic fertilizer on green maize plants under soil inoculation with Trichoderma harzianum and salinity stress? However, there are several works separately performed with organic fertilizer was performed combined to soil inoculation and salinity stress. In this study, however, the author examined all three factors together in terms of their response to drought stress in tomatoes. Here, the author claimed that a clear practical takeaway of 10 t ha⁻¹ manure + Trichoderma gave the best productivity under the tested conditions. This indicates real-world relevance with field-like variables (ECw, manure rates, inoculation). Below are some suggestions for improving clarity, accuracy and readability.

The work seems original and relevant to the field, but lacks sufficient novelty. Information on the Trichoderma strain/propagation method and the timing of inoculation, along with the reasons for it (L557), as well as the composition of the manure (C/N ratio and maturity) (L551), would facilitate replication. L373–375 and 387 appear contradictory. Although the study spans two seasons, it would be helpful to report year-by-year consistency and any related confounders. If possible, include data on ear/row counts, kernel weight and harvest index to pinpoint where gains occur. Given the practical recommendations, providing an approximate cost-benefit analysis or return on investment would enhance the potential for adoption. Please use precise terminology for salinity and irrigation (e.g. ECw values, soil EC and SAR, if relevant). Ensure that the units are consistent (t ha⁻¹ for manure and dS m⁻¹ for EC). I suggest including a detailed experimental matrix in table form: ECw levels, manure doses, inoculation versus non-inoculation and replication scheme. It would be great to have two- and three-way interaction plots to visualise how responses change across factors.

In addition, more details are needed on irrigation management, such as volume, leaching fraction and any rainfall events (based on the field). The author needs to provide a concise mechanistic discussion linking any observed physiological metrics to salinity mitigation mechanisms, such as osmolyte accumulation, ion balance and antioxidant activity. The author could also add a section on limitations and scope for future research, emphasising the need for field validation across soils and climates.

Author Response

Comment 1: The work seems original and relevant to the field, but lacks sufficient novelty. Information on the Trichoderma strain/propagation method and the timing of inoculation, along with the reasons for it (L557), as well as the composition of the manure (C/N ratio and maturity) (L551), would facilitate replication.

Response: The fungus Trichoderma harzianum (CEPA ESALQ 1306) was used in the research. It is a commercial product, the active ingredient in products such as Trichodermil, used in the research. The product comes in the form of a concentrated suspension with a concentration of 2.0 x 109 viable conidia/mL. It was applied according to the manufacturer's recommendation of 4L ha-1, calculated according to the experimental area. This value of 4L ha- was divided into 4 applications, the first on the day of sowing, and the others every 15 days, to ensure inoculation efficiency. Application was via soil (sprayed) according to the manufacturer's recommendation. Soil application is recommended because, in addition to influencing plant development, this product acts as a soil nematicide.

The manure used was cured, with advanced maturity and a C/N ratio of 18:1.

Comment 2: L373–375 and 387 appear contradictory. Although the study spans two seasons, it would be helpful to report year-by-year consistency and any related confounders. If possible, include data on ear/row counts, kernel weight and harvest index to pinpoint where gains occur.

Response: Suggestions accepted and added to the text of the article.

Comment 3: Given the practical recommendations, providing an approximate cost-benefit analysis or return on investment would enhance the potential for adoption.

Response: We appreciate the reviewer's suggestion to include an economic analysis. However, in this study, such an assessment was not feasible, as both the cattle manure and the maize seeds used were obtained at no cost; the manure was available in the experimental area itself, and the maize was donated by a local producer. Thus, it would not be possible to estimate costs or returns based on market values that do not reflect the actual conditions of the experiment.

Furthermore, the main objective of the study was to evaluate the agronomic, physiological, and productive responses of maize to inoculation with Trichoderma harzianum and to organic fertiliser doses. A meaningful economic analysis would require regional information and cost data under commercial conditions, which is beyond the scope of this research.

Comment 4: Please use precise terminology for salinity and irrigation (e.g. ECw values, soil EC and SAR, if relevant). Ensure that the units are consistent (t ha⁻¹ for manure and dS m⁻¹ for EC).

Response: Suggestions accepted and added to the text of the article.

Comment 5: I suggest including a detailed experimental matrix in table form: ECw levels, manure doses, inoculation versus non-inoculation and replication scheme. It would be great to have two- and three-way interaction plots to visualise how responses change across factors.

Response: Suggestions accepted and added to the text of the article.

Figure 12. Diagram of the experimental design showing the composition and interaction of the study factors — electrical conductivity of water, fertiliser doses and inoculation.

the figure was added to the article.

Comment 6: In addition, more details are needed on irrigation management, such as volume, leaching fraction and any rainfall events (based on the field). The author needs to provide a concise mechanistic discussion linking any observed physiological metrics to salinity mitigation mechanisms, such as osmolyte accumulation, ion balance and antioxidant activity. The author could also add a section on limitations and scope for future research, emphasising the need for field validation across soils and climates.

Response:

  1. In the first year of cultivation, the volume applied was 500 mm. In the second year of cultivation, it was 542 mm. Precipitation data are presented in the article.
  2. We recognise the importance of a more detailed mechanistic discussion of the possible physiological processes involved in mitigating salt stress. Although the present study did not include direct assessments of parameters such as osmolyte accumulation, ion balance, or antioxidant activity, the results obtained allow us to infer that the observed effects may be related to these mechanisms, as widely reported in the literature. Thus, although these mechanisms were not directly quantified in this study, it is plausible that the positive effects observed are related to similar physiological responses, which have been well documented in previous studies. This interpretation was included more clearly in the revised discussion.
  3. We have included a new section addressing the limitations of the study and perspectives for future research, highlighting the importance of validating the results in field conditions and under different soil and climate characteristics.

Reviewer 2 Report

Comments and Suggestions for Authors

The MS  "Organic fertilizer and biostimulant application to mitigate salinity stress in green maize under an agroecological system in  the Brazilian semiarid region" addresses important issues related to mitigating salt stress in plants. Unfortunately, it has many shortcomings.

1. In many cases, when differences are discussed, their significance is highly questionable. These differences are often within the error range. For instance on lines 317-318 the information is not true. There was no significant difference in plant height between 10 and 20 t/ha. Seasonal data also vary significantly, and in some cases, opposing trends are observed.

2. The title refers to salt stress. In fact, the plants show no signs of stress—leaf area, photosynthetic activity, and yield increase, while transpiration rates remain unchanged. This raises the question: were the plants experiencing stress?

3. There are inconsistencies in the reasoning. Lines 323-324 indicate that an increase in leaf area indicates enhanced tolerance, while lines 329-330 state that a reduction in leaf area is an adaptive mechanism for salt stress tolerance.

4. Lines 331-332 contain incorrect arguments regarding plant physiology. Water absorption by osmosis is a passive process and does not require energy.

5. The article title should indicate which organic fertilizer and biostimulant are being used.

6. In Fig. 1, the EC value in the first cycle should be indicated.

7. Trichoderma should be italicazed. 

Author Response

Comments 1:

In many cases, when differences are discussed, their significance is highly questionable. These differences are often within the error range. For instance on lines 317-318 the information is not true. There was no significant difference in plant height between 10 and 20 t/ha. Seasonal data also vary significantly, and in some cases, opposing trends are observed.

Response : The excerpt mentioned in lines 317–318 refers to results from another study, used only as a comparative reference. In the present study, in fact, no significant difference in plant height was observed between the doses of 10 and 20 t ha⁻¹, as shown in Figure 1. However, in the study by [21], used to contextualise the discussion, the authors reported a significant difference between these treatments, which justifies the mention made in the text.

Comments 2: The title refers to salt stress. In fact, the plants show no signs of stress—leaf area, photosynthetic activity, and yield increase, while transpiration rates remain unchanged. This raises the question: were the plants experiencing stress?

Response: Evaluating the work as a whole, it is possible to observe a reduction in some variables that were subjected to irrigation water with higher electrical conductivity, which suggests that the plants did indeed suffer stress. The work also suggests that the microorganism may have influenced the plant's response.

Comments 3: There are inconsistencies in the reasoning. Lines 323-324 indicate that an increase in leaf area indicates enhanced tolerance, while lines 329-330 state that a reduction in leaf area is an adaptive mechanism for salt stress tolerance.

Response: In this excerpt, ‘The reduction in leaf area under salinity stress is related to an adaptive mechanism by which plants minimise excessive water loss through transpiration,’ I referred to a general context regarding the results of several studies that point to the reduction of leaf area as an adjustment mechanism when plants suffer from salinity stress. In excerpts 323–324, I modified the text to avoid ambiguity, leaving: “The treatments with Trichoderma promoted a significant increase in leaf area, indicating that Trichoderma can reduce plant stress under saline conditions by stimulating antioxidant enzyme activity and gene expression […].

Comments 4: Lines 331-332 contain incorrect arguments regarding plant physiology. Water absorption by osmosis is a passive process and does not require energy.

Response : Actually, what I meant in lines 323 and 324 was: The reduction in leaf area under saline stress is related to an adaptive mechanism by which plants minimise excessive water loss through transpiration. This occurs because, under conditions of high salt concentration in the soil, the osmotic potential of the solution becomes more negative, hindering water absorption by the roots. In this situation, plants need to expend more energy on osmotic adjustment, accumulating compatible solutes in the cells to maintain a water gradient favourable to water entry.

Comments 5: The article title should indicate which organic fertilizer and biostimulant are being used.

Response: The suggested amendment was accepted.

Comments 6: In Fig. 1, the EC value in the first cycle should be indicated.

Response: Corrected in the text of the article.

Comments 7:  Trichoderma should be italicazed. 

Respoonse: The changes have been made.

Reviewer 3 Report

Comments and Suggestions for Authors

The manuscript titled "Organic fertilizer and biostimulant application to mitigate salinity stress in green maize under an agroecological system in the Brazilian semiarid region " contains interesting research results for both science and agricultural practice. 

I appreciate that this is a two-year field study. The text needs to be revised before publication. I have included detailed comments in the original PDF.

General comments:

Add: "salinity stress" to the keywords

Explain all abbreviations used below the tables.

If possible, improve the quality of the figures.

Check that you have presented all significant differences between the studied factors and their interactions in the figures.

Improve the description of Figure 10.

Line 316. Do not begin the sentence with [21]....
It is better to write "Eleduma et al. [21] reported."
Do the same throughout the text.

In the Materials and Methods sections, the references to the figures are incorrect. Correct references to figures and tables throughout the text.

Correct abbreviations and their descriptions in Table 4.

Briefly describe the results from Table 4.

Line 519. Provide the manufacturer, city, and country of the portable conductivity meter used.

Correct Table 5.

Provide the corn forecrop, tillage method, pesticide use, plot size, sowing dates, e.t.c

If manure was applied in divided doses, in what proportions (50/50).

Line 584. R4 - What is this plant development scale? Provide a source for literature of this scale

Correct the reference list according to the journal's requirements.

I hope my comments will help the authors improve the manuscript.

Author Response

Comments 1: Add: "salinity stress" to the Keywords

Response: As the expression “saline stress” appears in the title of the article, it was not mentioned in the keywords.

Comments 2: Explain all abbreviations used below the tables.

Response: Suggestion accepted and corrected in the article.

Comments 3: If possible, improve the quality of the figures.

Response: Suggestion accepted and corrected in the article.

Comments 4: Check that you have presented all significant differences between the studied factors and their interactions in the figures.

Response: All significant differences and their interactions were presented and reviewed.

Comments 5: Improve the description of Figure 10.

Response: Suggestion accepted and corrected in the article.

Comments 6: Line 316. Do not begin the sentence with [21]....
It is better to write "Eleduma et al.
[21] reported."
Do the same throughout the text.

Response: Suggestion accepted and corrected in the article.

Comments 7: In the Materials and Methods sections, the references to the figures are incorrect. Correct references to figures and tables throughout the text.

Response: Suggestion accepted and corrected in the article.

Comment 8:  Correct abbreviations and their descriptions in Table 4.

Response: Suggestion accepted and corrected in the article.

Comment 9: Briefly describe the results from Table 4.

Response: According to Table 4, the soil showed moderate fertility, with pH values ranging from slightly acidic to almost neutral (5.6–6.4) and an increase in organic matter and nutrient content in 2023 compared to 2022. There was also an increase in exchangeable calcium and sodium levels and a reduction in exchangeable magnesium and aluminium. In addition, there was a reduction in ECe in the 2023 cycle compared to 2022.

Comment 10: Line 519. Provide the manufacturer, city, and country of the portable conductivity meter used.

Response: Manufacturer: HM Digital; City: Not specified on the product packaging; Country: Designed in Korea and manufactured in China (Made in China).

Comment 11: Correct Table 5. Response - Corrected

Comment 12: Provide the corn forecrop, tillage method, pesticide use, plot size, sowing dates, e.t.c

Response: The experiment was conducted in an area of vegetation that was previously native. Before the experiment was set up, the area was not subjected to soil preparation and remained covered with native herbaceous/shrubby vegetation; therefore, there was no previous cultivation of maize in the experimental area. No pesticides, herbicides or fertilisers were applied to the experimental plot in the months prior to planting.

Comment 13: If manure was applied in divided doses, in what proportions (50/50).

Response : The manure was applied in two stages, the first before planting as a 30% foundation fertiliser. The remainder (70%) was applied after planting and establishment of the plants.

Comment 14: Line 584. R4 - What is this plant development scale? Provide a source for literature of this scale

Response: The plant development scale used was that proposed by Ritchie & Hanway (1993), which describes the phenological stages of maize (Zea mays L.) from emergence to physiological maturity. The stage referred to in the article as R4 corresponds to the dough stage.

Comment 15: Correct the reference list according to the journal's requirements.

Response: References have been corrected in accordance with the journal's guidelines.

Comment 16: I hope my comments will help the authors improve the manuscript.

We, the authors of this article, would like to thank you for all your suggestions and corrections. We believe that your suggestions enhance and improve the quality of our article. Thank you.

Round 2

Reviewer 1 Report

Comments and Suggestions for Authors

The limitations of the study and perspectives for future research, highlighting the importance of validating the results in field conditions and under different soil and climate characteristics need to adjust to the discussion and conclusion part.

Author Response

Comments 1: The limitations of the study and perspectives for future research, highlighting the importance of validating the results in field conditions and under different soil and climate characteristics need to adjust to the discussion and conclusion part.

Response: The suggestions were accepted and corrected in the text of the article. On behalf of all the authors, I thank you for your suggestions and corrections.

Reviewer 2 Report

Comments and Suggestions for Authors

The MS has been improved, but I don't agree with the phrase "plants must expend more energy to absorb water" (l. 374-375)

Trichoderma should be italicazed (l. 676-685).

Line 583 - In the beginning of the paragrapg - ... the volume applied ...  Please, specify what was applied. 

Author Response

Comments 1: The MS has been improved, but I don't agree with the phrase "plants must expend more energy to absorb water" (l. 374-375)

Response: This occurs because elevated salt concentrations decrease the osmotic potential of the soil solution, which restricts water movement into the roots.

Comments 2: Trichoderma should be italicazed (l. 676-685).

Response: The suggestions were accepted.

Comments 3: Line 583 - In the beginning of the paragrapg - ... the volume applied ...  Please, specify what was applied. 

Response: In the first year of cultivation, the volume applied was 500 mm of irrigation. In the second year of cultivation, it was 542 mm of irrigation.